# Gramiketides, Novel Polyketide Derivatives of *Fusarium graminearum*, Are Produced during the Infection of Wheat

**DOI:** 10.3390/jof8101030

**Published:** 2022-09-28

**Authors:** Bernhard Seidl, Katrin Rehak, Christoph Bueschl, Alexandra Parich, Raveevatoo Buathong, Bernhard Wolf, Maria Doppler, Rudolf Mitterbauer, Gerhard Adam, Netnapis Khewkhom, Gerlinde Wiesenberger, Rainer Schuhmacher

**Affiliations:** 1Department of Agrobiotechnology IFA-Tulln, Institute of Bioanalytics and Agro-Metabolomics, University of Natural Resources and Life Sciences Vienna (BOKU), Konrad-Lorenz-Str. 20, 3430 Tulln, Austria; 2Department of Applied Genetics and Cell Biology, Institute of Microbial Genetics, University of Natural Resources and Life Sciences Vienna (BOKU), Konrad-Lorenz-Str. 24, 3430 Tulln, Austria; 3Department of Plant Pathology, Faculty of Agriculture, Kasetsart University, Ngamwongwan Road, Lat Yao, Chatuchak, Bangkok 10900, Thailand; 4Core Facility Bioactive Molecules: Screening and Analysis, University of Natural Resources and Life Sciences Vienna (BOKU), Konrad-Lorenz-Str. 20, 3430 Tulln, Austria

**Keywords:** *Fusarium graminearum*, Fusarium head blight, wheat, *PKS15*, polyketides, gramiketide, LC-HRMS/MS

## Abstract

The plant pathogen *Fusarium graminearum* is a proficient producer of mycotoxins and other in part still unknown secondary metabolites, some of which might act as virulence factors on wheat. The *PKS15* gene is expressed only *in planta*, so far hampering the identification of an associated metabolite. Here we combined the activation of silent gene clusters by chromatin manipulation (*kmt6*) with blocking the metabolic flow into the competing biosynthesis of the two major mycotoxins deoxynivalenol and zearalenone. Using an untargeted metabolomics approach, two closely related metabolites were found in triple mutants (*kmt6 tri5 pks4,13)* deficient in production of the major mycotoxins deoxynivalenol and zearalenone, but not in strains with an additional deletion in *PKS15 (kmt6 tri5 pks4,13 pks15)*. Characterization of the metabolites, by LC-HRMS/MS in combination with a stable isotope-assisted tracer approach, revealed that they are likely hybrid polyketides comprising a polyketide part consisting of malonate-derived acetate units and a structurally deviating part. We propose the names gramiketide A and B for the two metabolites. In a biological experiment, both gramiketides were formed during infection of wheat ears with wild-type but not with *pks15* mutants. The formation of the two gramiketides during infection correlated with that of the well-known virulence factor deoxynivalenol, suggesting that they might play a role in virulence.

## 1. Introduction

*Fusarium graminearum* can infect roots, stem and flowering tissue of small grain cereals as well as maize and is the major causal agent of Fusarium head blight (FHB) on wheat and ear rot of maize [1]. The most prevalent mycotoxin produced by *F. graminearum* is deoxynivalenol (DON), which is not only toxic to humans and animals but has been demonstrated to be a major virulence factor in wheat. DON inhibits protein biosynthesis in infected host plants, thereby suppressing the synthesis of a mechanical barrier in the rachis of infected wheat ears and other defense mechanisms depending on pathogen induced proteins. *F. graminearum* is also a proficient producer of numerous other secondary metabolites [2,3].

A total of 76 predicted biosynthetic gene clusters (BGC), which are potentially involved in the production of these metabolites, have been reported so far [4]. Among those, 52 BGCs encode 16 putative polyketide synthases (PKS), 19 non-ribosomal peptide synthases (NRPS), and 17 terpenoid synthases (TPS) [2,4,5,6]. Secondary metabolite BGCs usually also contain additional genes encoding, for example, tailoring enzymes or cytochrome P450 (CYPs) for the metabolic modification of the core structures to the final secondary metabolites [2,4].

So far, the number of known *F. graminearum* metabolites is still limited. Of the fifteen PKS genes in strain PH-1 [4,6], only eight have been associated with the corresponding metabolites so far: A perithecium pigment (*PKS3*) [7], zearalenone (*PKS4* and *PKS13*) [8], gibepyrones and prolipyrone B (*PKS8*) [9], fusarielins (*PKS9*) [10], fusarin C (*PKS10*) [11], aurofusarin and rubrofusarin (*PKS12*) [12], and orsellinic acid and orcinol (*PKS14*) [13]. In addition, the fusaristatins have been reported to be associated with a combination of *PKS6* and *NRPS7* [10].

For a few of the secondary metabolites of *F. graminearum*, the role in infection and their effect on virulence of *F. graminearum* have been studied [14,15,16]. For example, different host-specific virulence factors were recently discovered. The cyclic lipopeptides gramillin A and B, products of NRPS8 in cluster C02 (according to Sieber et al. [2]), facilitate the fungus to promote cell death in maize leaves but do not affect wheat heads [15]. Likewise, the octapeptide fusaoctaxin A, derived from *NRPS5* and *NRPS9* genes from the C64 cluster, is expressed during wheat infection. Fusaoctaxin A helps the fungus to invade the wheat plant by cell-to-cell hyphal penetration [16]. Lately, a new cyclic hexapeptide, fusahexin, was identified in *NPRS4* overexpressing mutants. Overexpression, but not deletion of *NRPS4*, reduced virulence on wheat [17].

Molecular genetic methods are of great help in detecting and assigning so far unknown secondary metabolites, for which the expression of the underlying genes depends on very specific, often unknown, external factors. Epigenetic mechanisms can regulate gene expression through post-translational modifications targeting histones, mainly H3 and H4. The acetylation mark on lysine (K) residues is usually associated with loosely packed chromatin regions (i.e., euchromatin), resulting in easy access of the enzymatic transcription machinery and thus active gene transcription. In contrast, trimethylation of H3K9 and H3K27 to H3K9me3 and H3K27me3, for example, is associated with gene silencing by forming inaccessible heterochromatin. In *F. graminearum*, many secondary metabolite gene clusters are located in H3K27me3 enriched chromatin regions, and deletion of the *KMT6* gene, encoding the H3K27 methyltransferase, results in the derepression of many gene clusters involved in secondary metabolites production. Manipulation of repressive histone-modifying enzymes is suited to activate otherwise-silent BGCs that cannot be identified under standard culture conditions. However, only removing repressive histone modification does not guarantee the activation of silent BGCs. Generating mutations by many advanced strategies can be useful to reveal novel secondary metabolites and link them to the respective genes similar to more classic methods such as variation of cultivation parameters and homologous recombination of a constitutive promoter or a transcription factor gene for overexpression, and gene knock-out of negative regulators or competing enzymes [18]. For instance, a double mutant strain lacking *KMT6* and *FUS1* (*PKS10*, FGSG_07798, fusarin C biosynthesis) led to the identification of new metabolites, tricinolone and tricinolonic acid [4].

Gene clusters may be exclusively or more strongly expressed *in planta* under infection conditions compared to in vitro conditions. Moreover, co-expression of several (potentially related) BGCs can be observed during infection [2,7]. In *F. fujikuroi* for example, the fusaric acid gene cluster (*PKS6* and *NRPS34*) was coordinately upregulated at a high level during infection of maize together with the fusarin gene cluster (*PKS10*) [19]. In the case of *F. graminearum*, an earlier study reported that 28 gene clusters were highly expressed during plant infection, suggesting that they might play a role in pathogenicity or exhibit a host–pathogen-specific function. Interestingly, 20 of these BGCs had no associated metabolites assigned [2]. Among those, cluster C16 with the signature gene first named *PKS15* [20] at locus FGSG_04588 showed a notable expression profile with a significant increase at 96 h after infection by *F. graminearum*, indicating a possible role in virulence. Already in the first systematic analysis of the *F. graminearum* PKS genes [7], the lack of expression in vitro, but the presence of the transcript in wheat heads was noticed, and the *PKS15* gene (erroneously associated with gene FG04488) was named *PLSP1* (plant specific). Like many others, cluster C16 is also present in an H3K27me3 enriched chromosome region and showed a higher expression level in *KMT6* deficient mutants [18]. It should be mentioned that, in the latter publication, the *PKS15* gene was designated *PKS29*, following the nomenclature suggestion of Wiemann et al. [21]. Following authoritative reviews (e.g., [4,6]), we maintain the original designation of FGSG_04588 as *PKS15*.

Untargeted metabolomics in combination with genetic modifications of the fungal species under investigation has become an effective tool for the global, sensitive screening and characterization of novel secondary metabolites. In particular, stable isotope-assisted techniques, like the use of globally ^13^C, ^15^N, ^34^S or ^2^H labeled samples and the use of labeled tracer molecules under controlled experimental conditions, allow the identification and elucidation of biosynthetic pathways as well as the detection of novel compounds [22,23]. A comprehensive study of gene products and functions requires further isolation, purification, characterization, and structural elucidation of putative novel compounds using chromatographic techniques and NMR spectroscopy to link metabolites to the biosynthetic gene clusters and to fully elucidate their chemical structures.

Here, we successfully manipulated *F. graminearum* by generating a deregulated triple mutant (*kmt6 tri5 pks4,13*), deficient in production of the major mycotoxins DON and zearalenone (ZEN) but activating other gene clusters due to the deletion of *KMT6*. This led to the discovery of two candidate metabolites depending on an intact *PKS15* gene, which are in wild-type *Fusarium* only produced during infection of wheat. The tracer-assisted approach enabled the characterization of two previously unknown metabolites as polyketides and partial structure elucidation.

## 2. Materials and Methods

### 2.1. Generation of F. graminearum Mutant Strains

#### 2.1.1. Inactivation of the PKS15 Gene

Two plasmids were constructed using pKT248 [24]. An upstream fragment was amplified with primers 4232 and 4233 containing SpeI and SfiI sites (a list of primers used in this study is provided in Appendix A). Upon digestion with these enzymes, the fragment was cloned into SpeI and SfiI-digested plasmid pKT248 yielding plasmid pKR3 (see Appendix A for plasmids). A downstream fragment amplified with primers 4234 and 4235 was similarly cloned into pKT248 using HindIII and SalI (pKR9). Two restriction fragments containing the flanking regions and incomplete, overlapping fragments of the *hph* gene were used for transformation, namely a SfiI-AsiSI fragment from pKR3 and a HindIII-SacII from pKR9. The two fragments were mixed in an equimolar ratio to a total concentration of 10 µg DNA in 40 µL for *Fusarium* transformation.

#### 2.1.2. Generation of pks15 kmt6 Double Mutants

To generate the *pks15 kmt6* double mutants, we first constructed a plasmid allowing selection for G418 resistance, where the marker is flanked by *loxP* sites. A PCR fragment (*nptII* gene with *Aspergillus trpC* promoter and terminator) was generated using vector pII99 [25] as a template using primers 3151 and 3152. The amplified fragment was cleaved with SacI and XbaI (introduced by the primers) and cloned into the respective sites of pUG6 [26], resulting in pPS28 (*loxP-nptII-loxP*). Both the upstream and downstream flanking regions of *KMT6* were cloned into this vector, yielding pRS107. A linear NdeI-NotI fragment from this vector was used for transformation of PH-1. For *pks15 kmt6* double mutant generation, the split marker approach with fragments of pRS105 and pRS106 (which contain only one flank and a partial overlapping *nptII*) were used for transformation. pRS105 was digested with SfoI and NdeI, and pRS106 was cut with BcuI/PaeI and PvuI.

#### 2.1.3. Generation of a tri5 pks4,13 Double Mutant

To generate the *tri5 pks4,13* double mutant, first a hygromycin resistant transformant (dTRI5#11, Table 1) with disrupted *TRI5* gene (*tri5*∆::*loxP*-P*_XYN1_*-*Cre hph*-*loxP*, self-excising cassette) was selected and the marker subsequently removed on xylose medium. The resulting DON biosynthesis-deficient hygromycin-sensitive strain (dTRI5#11-19) was subsequently used to simultaneously delete both polyketide synthase genes, *PKS4* and *PKS13*, both of which are necessary for zearalenone biosynthesis [8]. To delete the two genes, which are transcribed in divergent orientations with a common promoter region, the flanking regions that correspond to the 3′ region of *PKS13* and the 3′ region of *PKS4* were amplified and cloned into separate vectors for subsequent use with the split marker approach.

#### 2.1.4. Complementation of the pks15 Mutation in the PH-1 Background

For the in-locus complementation of the *pks15* mutation in the PH-1 background (PPKS4), transformation was performed with a fragment containing the flanking regions and the *PKS15* open reading frame with a silent mutation.

We designed an additional HindIII site at position 7131 site via Fusion PCR (Appendix A). The triplet CTC encodes for Leucine. Changing the last base to CTT is synonymous but creates a HindIII site (AAG CTC → AAG CTT).

For Fusion PCR, two overlapping primers were designed:Forward Primer: 5′—CGTTAAGCTTCTGGAAGACGTGTAC—3′,Reverse primer: 5′—TACACGTCTTCCAGAAGCTTAACGGCGGCT—3′.

With these two primers creating the point mutation (underlined) and the primers for Gibson assembly F6, two fragments were amplified. In the next step, the two fragments were mixed in equimolar ratio and fused in a PCR using the F6-Gibson primers, yielding one fragment containing the additional HindIII site [27]. This fragment was then cloned via TOPO TA Cloning ^®^ (Invitrogen, Thermo Fischer Scientific, Vienna, Austria) into pKR91 using XbaI and MluI. The resulting plasmid, called pKR92, contains the coding region with the additional HindIII site between 5′ and 3′ flanking regions.

### 2.2. Cultivation of Fungi

#### 2.2.1. Cultivation in Liquid Minimal Medium

Submerged cultures were prepared for the genetically engineered strains since these strains do not require external stimulation but allow production of the target metabolites in synthetic minimal media, which greatly facilitates their detection. *Fusarium graminearum* PH-1 wild-type and knockout strains (two biological transformants each) of the mutants *kmt6*Δ::*nptII tri5*Δ::*loxP pks4*,*13*Δ::*loxP* (henceforth referred to as #1290 and #1291) and *kmt6*Δ::*nptII pks15*Δ::*loxP-hph-loxP tri5*Δ::*loxP pks4*,*13*Δ::*loxP* (henceforth referred to as #2263 and #2266, see Table 1) were grown in liquid minimal medium (1.0 g·L^−1^ KH_2_PO_4_, 0.5 g·L^−1^ MgSO_4_ 7 H_2_O, 0.5 g·L^−1^ KCl, 2.0 g·L^−1^ NaNO_3_, 10.0 mg·L^−1^ citric acid, 10.0 mg·L^−1^ ZnSO_4_·6 H_2_O, 2.0 mg·L^−1^ Fe(NH_4_)_2_(SO_4_)_2_·6 H_2_O, 0.5 mg·L^−1^ CuSO_4_ 5 H_2_O, 0.1 mg·L^−1^ MnSO_4_, 0.1 mg·L^−1^ H_3_BO_4_, 0.1 mg·L^−1^ Na_2_MoO_4_ 2 H_2_O all purchased from Sigma-Aldrich (Merck, Darmstadt, Germany), dissolved in ddH_2_O) containing 1% (*w*/*v*^−1^) D(+)-glucose as a sole carbon source. The cultivation was carried out in 24-well cell culture plates (TPP, Trasadingen, Switzerland) in a medium volume of 1.0 mL per well. In each case, 1 mL of sterile filtered medium (0.2 µm syringe cellulose acetate filters obtained from VWR) was inoculated with fresh spore solution (final concentration 4000 spores·mL^−1^) of the respective knockout strains. Six replicate biological cultivations were made per strain in a Memmert HPP260 (Memmert, Schwabach, Germany) incubator with the following conditions: constant darkness, 20 °C and 80% relative humidity. This protocol was applied in three variants as described in Section 2.2.2, Section 2.2.3 and Section 2.2.4.

#### 2.2.2. Cultivation on Native (^12^C) Glucose

For native cultivations, the protocol described in Section 2.2.1 was applied using D(+)-glucose (G8270 ≥ 99.5%) obtained from Sigma-Aldrich (St. Louis, MA, USA) as a sole carbon source.

#### 2.2.3. Cultivation on ^13^C Labeled Glucose

For the global labeling of the *F. graminearum* metabolome, U-^13^C_6_ D(+)-glucose obtained from Cambridge Isotope Laboratories, Inc. (Woburn, MA, USA) was used as a sole carbon source.

#### 2.2.4. Cultivation with Reversed Tracer Labeling Approach

To monitor the incorporation of C_2_ building blocks into the putative polyketides, the reversed tracer labeling approach described in Seidl et al. [28] was used. To this end, cultivation was carried out on U- ^13^C_6_ glucose culture medium (see Section 2.2.3), while native malonic acid monomethyl ester (Merck, Darmstadt, Germany) was used as metabolic tracer compound. (Upon hydrolysis of the methyl ester, malonic acid can be activated to malonyl-CoA and used for polyketide biosynthesis). The time point for the addition of the tracer (at the start of exponential growth) was determined by measurements of the glucose concentration in aliquots of the culture medium during the cultivation. For this purpose, 4 µL culture supernatant aliquots were taken every 24 h from each well, immediately after the culture plates had been shaken for 10 min at 140 rpm for homogenization. The sample aliquots were cooled on ice, quenched with 4 µL −20 °C cold ACN (Honeywell, Morristown, TN, USA) and centrifuged at 30,000× *g* for 10 min. The clear supernatant was transferred into HPLC vials with micro inserts and measured with HILIC-HRMS for monitoring glucose consumption. This was calculated by subtracting glucose concentrations of each timepoint from the concentration obtained 24 h ago. Exponential growth was assumed after glucose consumption had increased at least by a factor of four as compared to the (penultimate) consumption obtained 48 h ago. In addition, 1.0 mL of a 20 mM malonic acid monomethyl ester tracer solution was added to the cultures when exponential growth occurred. For strains #2263 and #2266, this was the case after 4–5 days and for the strains #1290 and #1291 after 6 days of pre-cultivation. After addition of the tracer solution, the culture plates were shaken at 140 rpm for 15 min before cultivation was continued. To determine the proper sampling time point, culture supernatant samples were taken every 48 h to further monitor glucose concentration as described above. The cultivations were stopped when the remaining glucose concentration reached 0.1% (*w*/*v*^−1^), corresponding to 10% of the initial concentration in the medium. This was the case after nine and 13 additional days of cultivation for the strains #2263, #2266 and #1290, #1291, respectively.

#### 2.2.5. Preparation of Fungal Samples for LC-HRMS Measurements

A large part of the fungal mycelium was removed using pipette tips and the supernatant thereafter filtered through glass wool to remove leftover mycelium. The filtered supernatant was then cooled on ice, and 700 µL of supernatant were quenched by adding 300 µL −20 °C cold ACN, to reach a final concentration of 30% (*v*/*v*^−1^) of organic solvent. Next, the samples were vortexed for 10 s and centrifuged at 30,000× *g* for 10 min at 4 °C. The supernatants were transferred into HPLC vials and immediately measured with LC-HRMS(/MS).

### 2.3. Cultivation and Treatment of Wheat Plants

#### 2.3.1. Plant Cultivation and Infection

Wheat seeds of the susceptible variety Remus were grown under controlled conditions in the glass house as described earlier [29]. At the stage of anthesis, ears were inoculated with *F. graminearum* spore solutions with a concentration of 4 × 10^4^ spores·mL^−1^ of the PH-1 wild-type strain as well as *pks15*Δ mutant and complemented mutant (*pks15 PKS15*-HindIII) strains.

For inoculation, 10 adjacent spikelets in the mid area of the wheat ear (two florets per spikelet) were treated by adding 10 µL aliquots of spore solution to the respective flower between palea and lemma. In total, 200 µL of spore solution were added per ear. Treatment with ddH_2_O (Mock treatment) served as control. Six replicates were made of each sample type. Immediately after inoculation, each ear was covered with a moistened plastic bag for the first 24 h to increase humidity and facilitate infection, and the plants were then further cultivated until sampling at 96 h after inoculation (hai).

#### 2.3.2. Preparation of Plant Samples

At 96 hai, the treated ears were cut off using clean and sterile scissors and were immediately frozen in liquid nitrogen. The frozen ears were ground for 30 s at 30 Hz in a Retsch ball mill in 35 mL grinding containers using a 20 mm steel ball (both pre-cooled in liquid nitrogen). In addition, 100 mg of milled, frozen material were then weighted into 2-mL Eppendorf tubes and stored in liquid nitrogen until extraction.

For extraction, ice-cold extraction solvent consisting of ACN/MeOH/H_2_O (1.5:1.5:1, *v*/*v*^−1^) including 0.1% formic acid (*v*/*v*^−1^) (Honeywell, Morristown, NJ, USA) was added to the samples in a ratio of 1 mL solvent per 100 mg sample. Samples were vortexed for 10 s, extracted (in an ice-cooled ultrasonic bath for 15 min) and centrifuged at 18,000× *g* for 15 min at 4 °C. In addition, 600-µL supernatant aliquots were transferred into new Eppendorf tubes. After addition of 300 µL H_2_O containing 0.1% formic acid (*v*/*v*^−1^), each tube was vortexed and centrifuged again as described above. Finally, the supernatant was transferred into HPLC vials and immediately measured with LC-HRMS(/MS).

### 2.4. LC-HRMS(/MS) Measurements

An Orbitrap QExactive HF (Thermo Fisher Scientific, San Jose, CA, USA) equipped with a heated electro-spray ionization source (HESI) coupled to a Vanquish UHPLC system (Thermo Fisher Scientific, San Jose, CA, USA) was used for separations of the sample constituents and data acquisition.

#### 2.4.1. Reversed Phase HPLC Method

For the measurements of fungal secondary metabolites, a reversed-phase (RP) XBridge BEH C_18_ column (150 × 2.1 mm i.d., 3.5 μm particle size, Waters, Milford, MA, USA) coupled with a pre-column (C_18_ 4 × 3 mm i.d., Security Guard Cartridge, Phenomenex, Torrance, CA, USA) were used. Operating conditions were as follows: The column temperature was set to 25 °C, and a sample injection volume of 2 µL was defined. Liquid phase was applied at a constant flow rate of 250 μL·min^−1^ and consisted of ELGA H_2_O with 0.1% formic acid (*v*/*v*^−1^) (eluent A) and MeOH with 0.1% formic acid (*v*/*v*^−1^) (eluent B). A gradient elution was applied starting with 90% A and 10% B for 2 min with a subsequent 30 min linear increase to 100% B followed by 100% B for 5 min and finally a re-equilibration at 10% B for 8 min (total run time 45 min).

#### 2.4.2. HILIC HPLC Method

For the monitoring of glucose, hydrophilic interaction chromatography (HILIC) with a SeQuant^®^ ZIC^®^-pHILIC column (100 × 2.1 mm i.d., 5 µm particle size, Merck Millipore, Burlington, MA, USA) was used. HPLC operating conditions were as follows: The column temperature and eluent pre-heater were set to 35 °C, and a sample injection volume of 1 µL was set. Liquid phase was applied at a constant flow rate of 300 μL·min^−1^ and consisted of 10 mM ammonium formate (Merk, Darmstadt, Germany) with pH 6 (Eluent A) and ACN with 5% ammonium formate (*v*/*v*^−1^) (eluent B). A gradient elution was applied starting with 100% B for 2 min with a subsequent 14 min linear decrease to 58% B (2 min hold time) and finally a 1-min linear gradient to 100% B (1 min hold time), resulting in a total run time of 20 min.

#### 2.4.3. High Resolution Mass Spectrometry (HRMS(/MS)) Settings

For Full-Scan MS^1^ measurements, the HESI was operated in fast polarity switching mode using the following MS and HESI parameter settings: Scan range: *m*/*z* 100–1500; Sheath gas flow rate 55 arb. units; Aux gas flow rate 5 arb. units with a temperature of 350°C, spray voltage 3.5 kV (positive mode) or 3.0 kV (negative mode); Resolving power R = 120,000 at *m*/*z* 200, AGC-target 3E6.

For data dependent analysis (DDA)-MS/MS measurements, the HESI was operated in positive ionization mode only, MS^1^ R = 120,000 and MS/MS R = 60,000 at *m*/*z* 200, CE stepped, 25/35/45 eV, isolation window 0.5 Da, MS/MS scans were only triggered by masses listed in an inclusion list containing the theoretical *m*/*z* values of the expected typical ESI-derived ion molecules, corresponding to X-n (*n* = 2, 4, 6, 8, 10, 12 and 14) isotopologs (with X being the fully ^13^C labeled form of the respective metabolite) as listed in Appendix A. All other parameters were equal to those of the Full-Scan MS^1^ measurements.

### 2.5. Data Evaluation (Screening and Structure Annotation)

#### 2.5.1. LC-HRMS Data Processing

For further data processing, the ThermoFisher LC-HRMS(/MS) raw files were converted into mzXML format using ProteoWizard MSConvert Software (version 3.0.19210 32-bit) [30] with the following settings: binary encoding precision 32-bit, writing index and TPP compatibility enabled, zlib and gzip compression disabled, and peak picking using a vendor algorithm. For the targeted data evaluation, XCMS (version 3.19.1) [31] and MSnbase (version 2.23.0) [32] were used. The MSnbase R-package was used to obtain basepeak chromatograms and extracted ion chromatograms (EICs) of the target features. The XCMS R-package was used for peak picking and for obtaining the peak areas. MS/MS spectra illustrations were also generated using XCMS whereby consensus spectra of all MS/MS scans acquired for the respective metabolite ion within the corresponding chromatographic peak were generated. To this end, the following settings were applied: intensity threshold 3 × 10^4^ counts, max tolerated *m*/*z* deviation of ± 3 ppm, and occurrence of a respective MS/MS peak in at least 85% all MS/MS spectra.

#### 2.5.2. Manual Metabolite Profile Screening

For the screening of differences between the metabolite profiles of the investigated *F. graminearum* strains, basepeak chromatograms of both positive and negative mode LC-MS^1^ data were generated, using *m*/*z* intervals of 50, respectively (i.e., 100–150 *m*/*z*, 150–200 *m*/*z,* etc.) and manually examined for qualitatively different peaks (i.e., present in strains, expected to produce the putative *PKS15* metabolite(s) but absent in the corresponding *pks15* mutant strains, see Table 1).

#### 2.5.3. MetExtract II Data Processing

For the MetExtract II (version 2.7.1) [33] data evaluation, 1:1 (*v*/*v*^−1^) mixtures of aliquots of each ^12^C culture supernatant with a pooled U-^13^C culture supernatant samples were prepared. Native ^12^C culture aliquots served as blanks (manual removal of potential false positives). The following settings of the AllExtract module were used for data processing: ^12^C isotope enrichment 98.93%, ^13^C isotope enrichment 99.51%, number of labeling atoms to search for 3–60, retention time window 3–36 min, intensity threshold 1000, charges ≤ 2, *m*/*z* deviation ± 3 ppm, required isotopologs (native and labeled) 2.

## 3. Results and Discussion

### 3.1. Genetics

Figure 1 gives an overview of the construction of the *F. graminearum* strains and their use in the present study for comparison of differences in the metabolite patterns.

In a first attempt, we tested whether differences exist in the metabolite profile between the wild-type strain PH-1 and *pks15* mutants. For this purpose, we constructed two plasmids (pKR3 and pKR9, see Materials and Methods) for generation of knockout mutants using the split marker technology. Two DNA fragments containing the flanking regions and incomplete overlapping fragments of the hygromycin phosphotransferase *hph* gene were used for transformation, and hygromycin resistant transformants generated by homologous recombination were screened for the desired gene knockout with flanking primers. Comparative evaluation of LC-HRMS total ion current and basepeak chromatograms between rice extracts of the PH-1 wild-type strain and two independent mutants (PPKS2 and PPKS3, Table 1) did not reveal any difference. This result was not unexpected, as it had been published that the *PKS15* gene is expressed only *in planta*, but not on an array of in vitro conditions [7].

As it had been reported that many secondary metabolite genes, including *PKS15* (designated *PKS29* in Connolly et al. [34]), are derepressed in a *kmt6* deletion strain, we also generated single *kmt6* and *pks15* double mutants. However, since *kmt6* mutants show poor growth, diminished conidiation and are hardly transformable, we inactivated *pks15* first (selecting for hygromycin resistance) and subsequently inactivated *KMT6* with *nptII* as a selection marker.

Using plasmid pRS107, which contains a loxP-*nptII*-loxP G418 resistance cassette, we transformed the wild-type strain PH-1 and obtained several PCR confirmed knockout mutants, two of which were selected for further experiments (IARS43, IARS45). To generate double mutants, we inactivated *KMT6* also in the *pks15* mutant strain PPKS4. Two PCR confirmed gene replacement mutants (KPKS5 and KPKS12) were used for subsequent metabolite analysis. However, no difference was found between the profiles of the *kmt6*Δ and of the *kmt6*Δ *pks15*Δ double mutants.

As a last resort, we hypothesized that maybe more of the *PKS15*-dependent metabolite might be formed if less of the flow of metabolic precursors is diverted to other competing compounds. We therefore generated *pks15* and *pks15 kmt6* mutants in a background that already had deletions of genes necessary for the first biosynthesis steps of the two major mycotoxins of *F. graminearum*, deoxynivalenol (DON) and zearalenone (ZEN). Strains with marker-less knockouts of *TRI5* and the two PKS genes needed for ZEN biosynthesis (*PKS4* and *PKS13*) were generated in a separate project, and their construction will be described elsewhere. In brief, self-excising cassettes consisting of two *loxP* sites flanking a hygromycin resistance gene and a fusion of the *Trichoderma reesei XYN1*-promoter with the *Cre* recombinase were used for disruption [35]. After selection of a hygromycin resistant transformant (dTRI5#11, Table 1) with a disrupted *TRI5* gene (*tri5*∆::*loxP*-(P*_XYN1_*-*Cre hph*)-*loxP*), the marker was removed, yielding a marker-less strain (*tri5∆::loxP*), which was subsequently used to delete both ZEN polyketide synthase genes [8] in one step. After pop-out of the reused hygromycin resistance, a markerless *tri5*Δ::*loxP pks4,13*Δ::*loxP* strain (IAG4_9_1, Table 1) was obtained. This background was then used to delete *PKS15* and to subsequently make the double mutant *pks15 kmt6*, finally resulting in the *F. graminearum* strains named KTPKS17 and KTPKS38 (#2263 and #2266 from now on), which were further used in this study (Table 1).

To validate that the metabolites found in vivo are indeed the metabolic end products of *PKS15*, we generated a complemented strain by transforming an engineered variant of the wild-type gene with a silent mutation introducing an additional HindIII site, which we subsequently used to verify the correct candidate. The transformant used in the metabolite analysis was selected by 5-FDU as previously described [24] (see also Appendix A).

### 3.2. Manual Screening for Differentially Produced Metabolites

To search for putative *PKS15* products, the metabolite profiles of *F. graminearum* wild-type (PH-1), the derepressed, i.e., *kmt6* deficient strains #1290 and #1291 and the *pks15* knockout strains #2263 and #2266 with the same genetic background were compared. A difference in the metabolite profile could only be observed under in vitro cultivation conditions with these triple mutants (*tri5 pks4,13 and kmt6*). The comparison of the metabolite profiles of the triple mutant and the additional *PKS15* knockout mutant then allowed two new metabolites associated with *PKS15* to be found. To this end, basepeak chromatograms of liquid culture supernatant samples were inspected and revealed two peaks in the positive ion mode basepeak chromatograms in the range of *m*/*z* 500–550, which were both found in the chromatograms of the “de-repressed” *kmt6* deficient strains #1290 and #1291 but were completely absent in those from the wild-type and *pks15* knockout strains (Figure 2). Based on these findings, the two underlying metabolites were considered to be the putative—so far unknown—*PKS15*-derived polyketides, which we had been looking for. We suggest naming the two target compounds gramiketide A and gramiketide B.

Analysis of the full scan mass spectra underneath the putative gramiketide A and gramiketide B peaks revealed [M + H]^+^ and [M + Na]^+^ adducts of the two candidate metabolites. They showed accurate *m*/*z* values at *m*/*z* 521.3109 ± 3 ppm and *m*/*z* 543.2928 ± 3 ppm at a retention time of 25.1 min, corresponding to the predicted sum formula C_29_H_44_O_8_ for gramiketide A and *m*/*z* values at *m*/*z* 503.3003 ± 3 ppm and *m*/*z* 525.2823 ± 3 ppm at 28.4 min, corresponding to the presumable sum formula C_29_H_42_O_7_ for gramiketide B. No corresponding EIC peaks or *m*/*z* signals were found in the negative ionization mode, neither for gramiketide A nor gramiketide B.

The initial target compounds were confirmed by a global ^13^C labeling approach and MetExtract II data evaluation as described in [33]. With this approach, pairs of native and U-^13^C labeled isotopologs were detected at the same retention times, demonstrating that both target metabolites were truly produced by *F. graminearum*. In addition, the observed mass increments between the two major isotopologs corresponded to 29 carbon atoms per formula unit, thereby supporting the molecular formulas predicted from the accurate mass and natural isotope patterns (Figure 3).

### 3.3. Structural Analysis by Reversed Isotopic Labeling and LC-HRMS/MS

To further characterize the structure class of the two target metabolites gramiketide A and B, malonic acid monomethyl ester feeding trials were carried out. The reversed labeling tracer approach showed the specific isotopolog pattern that was expected for the iterative, random incorporation of either labeled or tracer-derived C_2_ extension units, resulting in isotopologs with an *m*/*z* shift with multiples of 2.00671 Da and stepwise decreasing peak intensities with increasing numbers of incorporated tracer-derived C_2_ extension units. The observed characteristic isotopolog pattern proves that native C_2_ units were incorporated into the carbon backbone of the target molecules after cultivating *F. graminearum* on U-^13^C_6_ glucose in the presence of unlabeled ^12^C malonic acid monomethyl ester as a tracer. These findings are in good agreement with the biosynthesis pathway of polyketides, which involves a stepwise extension of the growing polyketide chain by Claisen condensation reactions. Chain extensions are catalyzed by the ketosynthase domain of the PKS connecting the cysteine-bound polyketide intermediate with a malonyl extender unit under cleavage of carbon dioxide from the extender unit. A closer inspection of the resulting isotopolog patterns indicated the overlay of two separate patterns, evident from the clearly observed interruption of the stepwise, continuous intensity decrease after the isotopolog with four tracer-derived extension units. This is not the expected pattern for a polyketide backbone, which is synthesized in one stretch by a single PKS enzyme. The observed superimposition of two tracer-evoked stepwise falling isotopolog patterns suggests that the tracer is used to produce two distinct biosynthetic intermediates that are presumably assembled to the final biosynthetic product. Similar isotopolog patterns were found for both metabolites (Figure 4).

In both cases, the incorporation of at least 10 C_2_ units originating from the tracer into the fully labeled target compounds can be seen in the isotopolog patterns, i.e., 20 carbon atoms originating from the labeled malonate (Figure 3). Beyond 10 C_2_ units, the MS signal gets too low to determine whether further extension by malonate-derived acetate units occurred or whether the remaining carbon atoms no longer originate from the tracer moiety. The sum formulas of the two ketide-derivatives differ by one oxygen and two hydrogen atoms, suggesting that the chemical structures of the two compounds are closely related to each other. Indeed, the LC-HRMS/MS product ion spectra of gramiketide A and gramiketide B are very similar and share all major fragments. The only significant difference originates from the loss of a water molecule, which is well in line with the assumption that both compounds are produced by one PKS and share the same core structure. However, the two compounds might still differ in their stereochemical configuration that might not be detected by MS/MS (Figure 5).

A list of all peaks in the MS/MS spectrum of gramiketide A with a relative intensity > 10% are listed in Table 2, along with annotations of elementary composition of the fragment and neutral loss and theoretical *m*/*z* values of the respective fully _13_C labeled fragments.

The applied reversed labeling approach allows for obtaining further structure information by comparison of MS/MS spectra of the monoisotopic U-^13^C_29_ precursor (no tracer incorporated), to those of partly tracer-enriched analogues (selected precursors are listed in Appendix A). Mass increments of corresponding molecules/fragments directly indicate the number of tracer moieties contained in the respective ion species. In the example given in Figure 6, the partly labeled intact protonated molecule contains six tracer-derived C_2_ extension units as can be seen from the mass shift of −12.04026 Da.

While in the two heavier fragments (labeled (1) and (2)), the carbon skeleton is still fully intact (loss of water), fragment (3) results from an additional loss of a CH_2_ unit. However, as no splitting of mass signals in the labeled fragment is observed (as would be expected from the fact that a given number of tracer units, six in the illustrated example, can be incorporated at different sites of the molecule), it can be concluded that the carbon atom of the cleaved CH_2_ moiety is not derived from the labeled tracer. In contrast, the fragment ions at *m*/*z* 221.0943 (4), 206.0753 (5) and 188.0647 (6) corresponding to ^13^C_10_H_11_O_5_, ^13^C_9_H_9_O_5_ and ^13^C_9_H_7_O_4_ respectively clearly show multiple isotopologs demonstrating that at least three C_2_ extension units of these fragments originate from the tracer.

Interestingly, the fragment ion with m/z 120.1195, corresponding to ^13^C_7_H_13_O, is shared between the fully ^13^C labeled and its partly labeled analogue and does not show any signal splitting. This means that this fragment does probably not contain any tracer-derived C_2_ units and might therefore not represent a polyketide but a structurally different moiety that was biosynthetically assembled with the polyketide part to form the final metabolite. This is well in line with the above-described labeling patterns of the intact gramiketides, which also suggest a composite structure of two distinct parts of different biosynthetic origin. This observation is also supported by the gene annotation of the *F. graminearum* C16/*PKS15* gene cluster (see Appendix A).

In addition to a polyketide synthase (FGSG_04588), the cluster also contains a putative farnesyltransferase (FGSG_04591) as well as two methyltransferases which have been shown to be significantly upregulated 72 h after infection of barley [2]. It is therefore likely that the gene product is not a pure polyketide, but a hybrid consisting of a polyketide- and e.g., terpene part, potentially connected by a transferase that is also annotated in the cluster. Interestingly, an evaluation of the LC-HRMS/MS spectra of both gramiketides using CANOPUS (Sirius software, version 5.5.7) [36,37,38,39] resulted in the prediction of the natural substance classes for both metabolites to be meroterpenoids and the corresponding metabolic pathways polyketides and terpenoids. This result also fits in well with the overall picture that we have gained of the two new metabolites so far. For a more detailed characterization of the gramiketide structures, also further tracer substances like, for example mevalonate, a metabolic precursor of terpenoids in fungi could be used in the future. Since mass spectrometry ultimately is not sufficient for a complete structural elucidation, we also tried to purify and analyze the supernatant samples from the strains #1290/91 (triple mutant) cultivated on minimal medium by ^1^H NMR and ^13^C NMR. However, the quantity of target substance produced under these conditions, even after scale-up to 5 L batches, was not sufficient to generate meaningful NMR spectra. We then started an attempt to extract and isolate the metabolites from the wild-type infected wheat ears (14 dai, Section 3.4). While here, the amount of the two metabolites obtained seemed adequate, we have not yet achieved sufficient purification from the very complex plant matrix by means of preparative HPLC. Alternative production, purification, and identification approaches are beyond the scope of this study but will be followed in the near future.

### 3.4. Merge of Results from In Vitro Cultivations and Plant Experiments

Earlier publications on FHB reported that the biosynthetic *PKS15* cluster of *F. graminearum* is only expressed in planta under infection conditions in barley and wheat [2] but not in vitro under lab culture conditions [7]. Based on these findings, we infected wheat with different strains of *F. graminearum* with the aim to unambiguously demonstrate that the putative target compounds, selected from the differential comparison of *F. graminearum* strains #1290/91 and #2263/66 are the true products of the *PKS15* derived enzyme. For this purpose, flowering wheat ears were inoculated with spores of *F. graminearum* wild-type strain PH-1 as well as with the *pks15*Δ and the complemented *pks15* mutant strains for 96 h and analyzed by LC-HRMS/MS. A targeted search for the putative gramiketides A and B revealed that the identical target metabolites were produced in plants infected with both the PH-1 wild-type and the complemented strains (both containing a functional *PKS15* gene), but not after treatment with the *pks15*Δ strain (Figure 7).

In addition to the matching accurate masses and retention times, LC-HRMS/MS spectra of *F. graminearum* in vitro culture- and pathogen infected wheat samples were compared. With the dot product [40] of 0.984, a high similarity score was achieved, further supporting that not only are the gramiketides A and B true products of the *PKS15* cluster but also that the polyketide derivatives produced by the fungus in vitro are identical to those produced under infection conditions *in planta* (Figure 8).

### 3.5. Time Course of Metabolite Abundance in Planta

To investigate the formation of the gramiketides A and B *in planta* over time, LC-HRMS/MS data of an earlier time course experiment in the glasshouse were re-evaluated, in which flowering wheat ears were infected with spores of *F. graminearum* wild-type (PH-1) and sampled at different time points up to 96 h (five repetitions per time point). For details of the biological experiment, refer to Schweiger et al. [41]. As expected from the results described in the literature [2,42,43,44], the target metabolites were formed upon infection of wheat and were measurable with the described RP-LC-HRMS method 72 h after treatment, with concentrations further increasing over time (Figure 9).

The time course profiles of gramiketide A and B correlated well with the transcription level of the *PKS15* gene FGSG_04588 over time. In addition, the measured relative abundances of the dominant gramiketide B also correlated well with those of the well-known virulence factor DON in the same samples. In this context, it should be noted that MS/MS spectra comparisons of gramiketides with deoxynivalonol and zearalenone have not shown structural similarities (Appendix A). Together with the fact that the *PKS15* cluster is only expressed under infection conditions *in planta*, our study suggests an active role of gramiketides in the infection process by *F. graminearum*. Interestingly, a number of further, only recently discovered pathogen-derived secondary metabolites like the NRPS5/NRPS9 produced nonribosomal octapeptide fusaoctaxin A [16,45,46] and the NRPS8-derived gramillins [15] were both found to act as virulence factors of *F. graminearum* in wheat and maize, respectively. While fusaoctaxin A facilitates cell to cell invasion by the pathogen mainly by inhibiting the closure of plasodesmata and formation of cell wall depositions in wheat [16], the gramillins are produced by *F. graminearum* on maize silks, where they presumably induce cell death in host tissue during colonization [15]. It will be interesting to further study the occurrence of the gramiketides in other cereals and their role during infection of host plants.

## 4. Conclusions

Fusarium head blight (FHB) is a relevant disease of wheat worldwide, causing dramatic crop losses and large-scale mycotoxin contamination every year. FHB is a highly complex disease with many fungal attack and plant defense strategies involved. Despite years of research, FHB disease is still not fully understood, not least due to the fact, that still not all substances and strategies involved during attack and disease development are known. In order to get a more holistic picture of the devastating FHB disease, and thus to be able to take better action against it, it is essential to track down and examine all the systems and natural compounds implicated. In this work, we have successfully amended the strategy to manipulate the fungal chromatin structure with the novel approach to diminish the metabolic flow into competing metabolic pathways.

Here we identified two new candidate metabolites, named gramiketide A and B, associated with the polyketide cluster *PKS15*, which is specifically expressed during infection by wild-type *F. graminearum*. While we failed to obtain sufficient pure metabolites to allow unambiguous structure determination by NMR, the mass spectroscopic evidence obtained strongly suggests that the product of the gene cluster is a terpene-modified polyketide. Its accumulation pattern early in infection suggests that the identified gramiketides might be relevant for overcoming the defense mechanisms of wheat during *F. graminearum* infection. Based on this, it seems warranted to re-evaluate the former findings by Gaffoor et al. [7], who reported that *PKS15* deletion mutants remained pathogenic on inoculated wheat heads. Further investigation of the structure of the *PKS15*-associated compounds, their mode of action *in planta*, and the role in the infection of wheat in future work seems rewarding.

## Figures and Tables

**Figure 1 jof-08-01030-f001:**
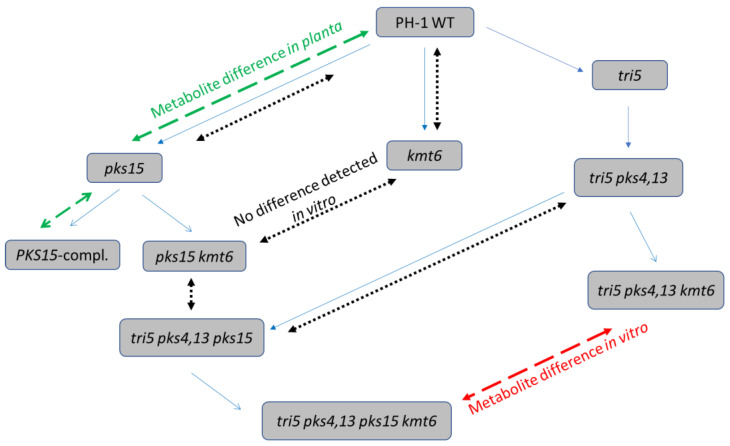
Overview of the generation of *F. graminearum* knockout strains and observed metabolite differences. Blue arrows indicate the strain genealogy (order of gene disruptions). Dashed arrows indicate the results of metabolite comparisons (Red: difference in vitro. Green: difference *in planta*. Black: no difference in vitro detected).

**Figure 2 jof-08-01030-f002:**
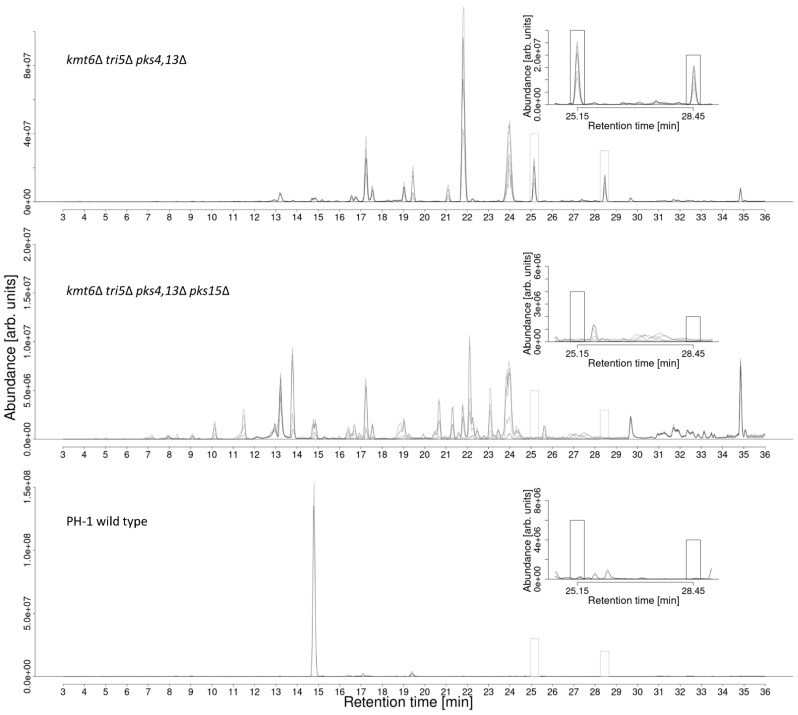
Comparison of basepeak chromatograms with *m*/*z* range 500–550 of de-repressed *kmt6*Δ strains #1290 and #1291 (top chromatogram) *kmt6*Δ *pks15*Δ knockout strains #2263 and #2266 (middle chromatogram) and PH-1 wild-type (bottom chromatogram) culture supernatant samples. Two additional peaks marked with boxes were found at retention time 25.1 min and 28.4 min, respectively. These were exclusively present in the chromatograms of the strains #1290 and #1291 but missing in those of the *pks15*Δ and wild-type strains (on minimal media).

**Figure 3 jof-08-01030-f003:**
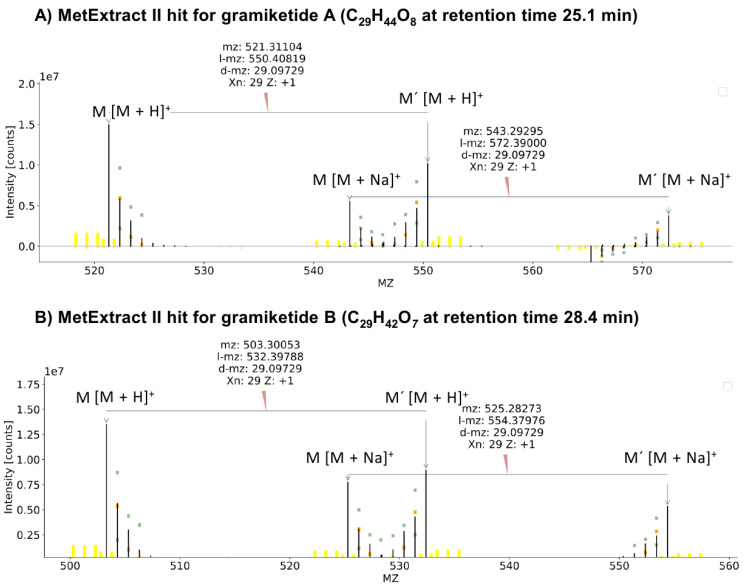
MS^1^ spectra of metabolite ions, identified by MetExtract II in the native/uniformly ^13^C labeled 1 + 1 mixed supernatant samples of strains #1290 and #1291 after cultivation in liquid minimal medium. (**A**) the spectrum for [M + H]^+^ and [M + Na]^+^ ions of C_29_H_44_O_8_ and (**B**) the spectrum for [M + H]^+^ and [M + Na]^+^ ions of C_29_H_42_O_7_, confirming the number of carbon atoms contained to be 29 in both cases.

**Figure 4 jof-08-01030-f004:**
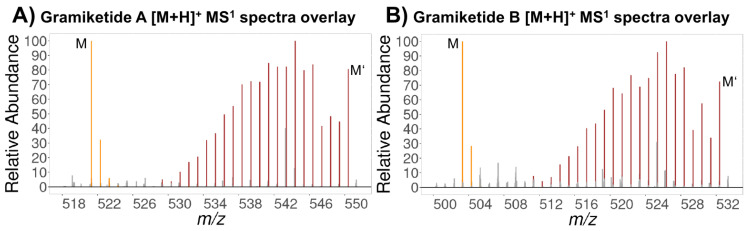
Average MS^1^ [M + H]^+^ ion spectra of gramiketide A (**A**) and gramiketide B (**B**) each showing an overlay of the native ^12^C isotopolog pattern (orange) and ^13^C labeled isotopologs with a varying number of ^12^C tracer units incorporated (brown). Unrelated signals are shown in gray. Peak intensities were normalized to most abundant signals of ^12^C and ^13^C isotopologs, respectively.

**Figure 5 jof-08-01030-f005:**
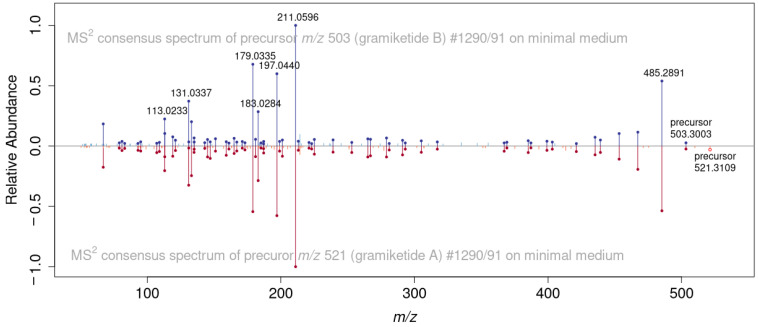
Comparison of MS/MS spectra of [M + H]^+^ precursor ions of gramiketide A (red) and gramiketide B (blue), demonstrating the close structural similarity of the two metabolites. Matching fragments are marked with dots.

**Figure 6 jof-08-01030-f006:**
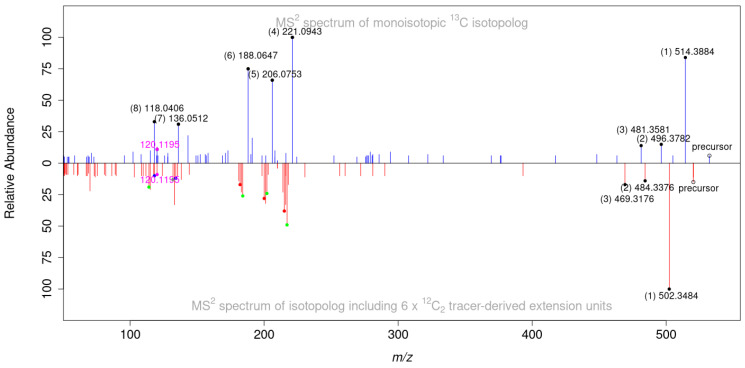
Comparison of MS/MS spectra of the fully ^13^C labeled form of C_29_H_42_O_7_ [M + H]^+^ ion at *m*/*z* 532.3976 (shown in blue) with the corresponding isotopolog carrying six tracer-derived extension units in the metabolite (shown in red). Peaks with corresponding *m*/*z* shifts are marked with black dots in the monoisotopic spectrum and as colored dots in the tracer containing isotopolog, indicating the number of tracer-derived extention units (from 6 to 3 extension units, marked as black (6), red (5), green (4) and blue (3), respectively). A single fragment of identical mass not showing a mass shift is marked in magenta. Numbers in brackets indicate the respective peak assignment.

**Figure 7 jof-08-01030-f007:**
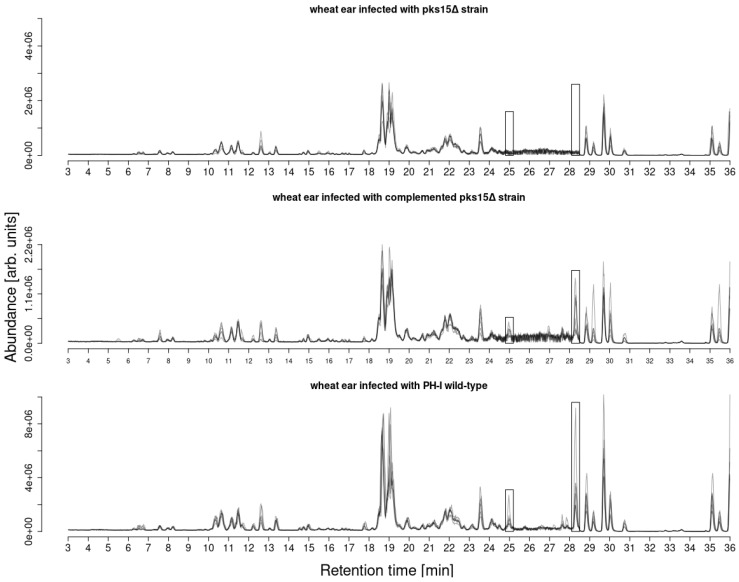
Comparison of basepeak chromatograms (*m*/*z* range 500–550) of wheat ear extracts of Remus plants treated with *F. graminearum* complemented *pks15*Δ (**upper** panel), *pks15*Δ (**middle** panel) and PH-1 wild-type (**lower** panel) strains. Gramiketides A and B marked in boxes. Both gramiketides were found to be present in the complemented *pks15* mutant and wild-type PH-1 treated wheat samples but missing in the *pks15*Δ treated samples.

**Figure 8 jof-08-01030-f008:**
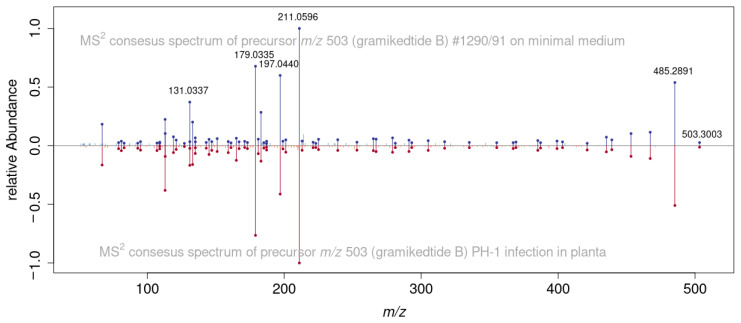
Comparison of LC-HRMS/MS spectra of gramiketide B *m*/*z* 503.3003 produced by the *kmt6*∆ strain in liquid minimal medium (blue) and the metabolite produced by PH-1 in planta (red) showing a high similarity (dot product similarity score of 0.984) of the spectra.

**Figure 9 jof-08-01030-f009:**
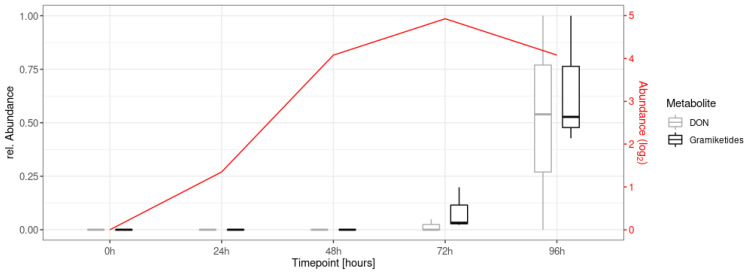
Time course measurement of gramiketide B and deoxynivalenol (DON) abundances from samples taken immediately (0 h) to four days after infection (point inoculation of wheat flowers with *F. graminearum* PH-1 wild-type at anthesis). Peak areas are normalized relative to the highest value per metabolite. Transcription levels (for experimental details and methods of obtaining transcription data, refer to Schweiger et al. [41]) of the *PKS15* gene (FGSG_04588) are shown in red.

**Table 1 jof-08-01030-t001:** Construction and genotypes of *Fusarium* strains used in this study.

Strain Name	Construction (Plasmids Used for Transformation of Parental Strain)	Genotype	Strain Number
PH-1	Wild-type (NRRL31084)	-	
PPKS2	PH-1/pKR3 + pKR9	*pks15*∆::*loxP*-(P*_PKI1_HSV-TK hph*T*cbh2*)-*loxP*	1983
PPKS4	PH-1/pKR3+pKR9 (independent transformant)	*pks15*∆::*loxP*-(P*_PKI1_HSV-TK hph*T*cbh2*)-*loxP*	1984
IARS43	PH-1/pRS107	*kmt6*∆::*loxP*-(P*_trpC_ nptII* T*_trpC_*)-*loxP*	1288
IARS45	PH-1/pRS107 (independent transformant)	*kmt6*∆::*loxP*-(P*_trpC_ nptII* T*_trpC_*)-*loxP*	1289
KPKS5	PPKS4/pRS105+pRS106	*pks15*∆::*loxP*-(P*_PKI1_HSV-TK hph*T*cbh2*)-*loxP kmt6*∆::*loxP*-(P*_trpC_ nptII* T*_trpC_*)-*loxP*	2025
KPKS12	PPKS4/pRS105+pRS106	*pks15*∆::*loxP*-(P*_PKI1_HSV-TK hph*T*cbh2*)-*loxP kmt6*∆::loxP-(P*_trpC_ nptII* T*_trpC_*)-loxP	2026
dTRI5#11	PH-1 split marker 3.6 kb NotI-PstI from pGW859-12 (486 bp *TRI5* 5′ region with up-tag Not-SpeI in pGW859), 2.1 kb NotI-SacII from pGW860-19 (503 bp *TRI5* 3’ region with down-tag in pGW851)	*tri5*Δ::*loxP*-(*hph* P*_XYN1_*-*Cre*)-*loxP*	330
dTRI5#11-19	dTRI5#11 Cre-mediated popout (hygromycin sensitive)	*tri5*Δ::*loxP*	461
IAG4_9	dTRI5#11-19, split hph - *PKS4* and *PKS13* flanking regions	*tri5*Δ::*loxP pks4,13:: loxP*-(*hph* P*_XYN1_*-*Cre*)-*loxP*	614
IAG4_9_1	IAG4_9 Cre-mediated popout, hygromycin sensitive	*tri5*Δ::*loxP pks4,13:: loxP*	750
**IARS49**	IAG4_9_1/pRS105+pRS106	*tri5*Δ::*loxP pks4,13:: loxP kmt6*∆::loxP-(P*_trpC_ nptII* T*_trpC_*)-loxP	**1290**
**IARS54**	IAG4_9_1/pRS105+pRS106	*tri5*Δ::*loxP pks4,13:: loxP kmt6*∆::loxP-(P*_trpC_ nptII* T*_trpC_*)-loxP	**1291**
TPKS2	IAG4_9_1/pKR3+pKR9/popout	*tri5*Δ::*loxP pks4,13:: loxP* pks15∆::loxP	2119
TPKS4	IAG4_9_1/pKR3+pKR9/popout	*tri5*Δ::*loxP pks4,13:: loxP* pks15∆::loxP	2120
KTPKS17	TPKS4/pRS105+pRS106	*tri5*Δ::*loxP pks4,13:: loxP pks15*∆::*loxP*-(P*_PKI1_HSV-TK hph*T*cbh2*)-*loxP kmt6*∆::*loxP*-(P*_trpC_ nptII* T*_trpC_*)-*loxP*	2263
KTPKS38	TPKS4/ pRS105+pRS106	*tri5*Δ::*loxP pks4,13:: loxP pks15*∆::*loxP*-(P*_PKI1_HSV-TK hph*T*cbh2*)-*loxP kmt6*∆::*loxP*-(P*_trpC_ nptII* T*_trpC_*)-*loxP*	2266
7C PPKS3	PPKS4 in locus complemented/additional silent HindIII in ORF	*PKS15-C* *7131T*	2284

**Table 2 jof-08-01030-t002:** MS/MS fragment annotations of gramiketide A spectrum.

	*m*/*z* (Native)	Rel. Abundance	Formula	Formal Loss	*m*/*z* Fully ^13^C
Precursor	521.30902	0.01	C_29_H_45_O_8_	-	549.40291
Fragment 1	485.28986	0.51	C_29_H_41_O_6_	H_4_O_2_	514.38706
Fragment 2	467.27933	0.22	C_29_H_39_O_5_	H_6_O_3_	496.37649
Fragment 3	453.26367	0.11	C_28_H_37_O_5_	CH_8_O_3_	481.35749
Fragment 4	267.12268	0.10	C_14_H_19_O_5_	C_15_H_26_O_3_	281.16967
Fragment 5	211.06017	1.00	C_10_H_11_O_5_	C_19_H_34_O_3_	221.09365
Fragment 6	197.04457	0.53	C_9_H_9_O_5_	C_20_H_36_O_3_	206.07464
Fragment 7	183.02893	0.22	C_8_H_7_O_5_	C_21_H_38_O_3_	191.05564
Fragment 8	179.03395	0.71	C_9_H_7_O_4_	C_20_H_38_O_4_	188.06408
Fragment 9	165.01833	0.13	C_8_H_5_O_4_	C_21_H_40_O_4_	173.04507
Fragment 10	145.1013	0.11	C_11_H_13_	C_18_H_32_O_8_	156.13808
Fragment 11	133.10135	0.26	C10H13	C_19_H_32_O_8_	143.13473
Fragment 12	131.03407	0.18	C_5_H_7_O_4_	C_24_H_38_O_4_	136.05066
Fragment 13	113.02368	0.36	C_5_H_5_O_3_	C_24_H_40_O_5_	118.04009
Fragment 14	67.01859	0.22	C_4_H_3_O	C_25_H_42_O_7_	71.03126

## Data Availability

Data are contained within the article or Appendix A.

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
