# Peer review of "Gramiketides, Novel Polyketide Derivatives of Fusarium graminearum, Are Produced during the Infection of Wheat"

_jof, 2022, doi:10.3390/jof8101030_

Round 1

Reviewer 1 Report

This paper describes the identification of two new candidate metabolites from F. graminearum, which the authors have termed as gramiketide A and B, - associated with the polyketide cluster PKS15. In fact, to establish the hypothesis, the group has constructed a series of mutants of F. graminearum PH-1 strain (wild type). Strategies for mutagenesis, counterselection, application and analysis of metabolites are well defined and explained. The manuscript has been nicely prepared and remain focused on the primary objective of the work. The technical aspect of identification of metabolite using stable isotope-assisted tracer approach is also justified. It may be accepted for publication.

There are few very minor observations, which may be considered.

1. Correct - The PKS15 gene of Fusarium graminearum is expressed only in planta, so… line  Also, this fact need to be elaborated in Introduction.

2. …..of kmt6 tri5 pks4,13 strains depending on PKS15…. Line 23, .. In the abstract, without the reference/significance of these strains, the readers may not get the idea of work. The general message may be explained. (a triple mutant deficient in production of the major mycotoxins….)

3. line 23-27, long and confusing statement… rephrase. (…. for which we suggest the names gramiketide A and B, by LC-HRMS/MS in…- need attention).

4. For a few of the secondary F. graminearum metabolites…. Change to ‘For a few of the secondary metabolites of F. graminearum’ line 59

5. Line 115-120: What is the recent nomenclature of PKS15? It has been used in this paper elsewhere too, or PKS29 being more recent, should be used?

6. Section 2.2.2 and 2.2.3, need not to be separated with 2.2.4. may be described together.

7. line 271, 4*104, replace with 4x104

Reviewer 2 Report

The manuscript entitled (Gramiketides, novel polyketide derivatives of Fusarium graminearum are produced during the infection of wheat) by Seidl et al. reported the identification of two gramiketides A and B, novel polyketide derivatives of Fusarium graminearum are produced during the infection of wheat (FHB) by LC-HRMS/MS.

The manuscript could be accepted after covering the following issues

1-    In keywords, please correct wheat to (Wheat) and add polyketides, LC-HRMS/MS.

2-    The introduction is too long, the author should summarize it.

3-    The manuscript contains many abbreviations, so please add a list of all abbreviations at the end of manuscript.

4-    Are the two gramiketides A and B mycotoxins?. Did the author try to separate them from the fraction/s that contain these compounds by using any technique as HPLC and testing them. This will be important to highlight and avoid the deleterious effect of this fungus on human health.

5-    Are the compounds related to deoxynivalenol or zearalenone in the structures?

6-    The authors calculated molecular weight and give a molecular formula for both compounds, so, what are the suggested structures of gramiketides A and B based on their fragmentation patterns? Also, what are the expected subunits for each fragment?

7- In figure 2, resolution needs improvement.

8-    My suggestion is that the authors should cultivate the fungus on a large scale to get a sufficient amount of these metabolites to establish their structures by NMR. I think the MS/MS is not sufficient for the characterization of new natural products.

Round 2

Reviewer 2 Report

No comments